# Reweighting and Reembedding are needed for Tail-item Sequential Recommendation

## ABSTRACT

Large vision models (LVMs) and large language models (LLMs) are becoming cutting-edge for sequential recommendation, given their success in broad applications. Despite their advantages over traditional approaches, these models suffer more significant performance degradation on tail items against conventional ID-based solutions, which are largely overlooked by recent research. In this paper, we substantiate the above challenges as (1) *all-in ground-truth*, *i.e.,* the standard cross-entropy (CE) loss focuses solely on the target items while treating all non-ground-truth equally, causing insufficient optimization for tail items, and (2) *knowledge transfer tax*, *i.e.,* the knowledge encapsulated in LLMs and LVMs dominates the optimization process due to insufficient training for tail items. We propose *reweighting and reembedding*, a simple yet efficient method to address the above challenges. Specifically, we reinitialize tail item embedding via a Gaussian distribution to alleviate knowledge transfer tax; besides, a reweighting function is incorporated in the CE loss, which adaptively adjusts item weights during training to encourage the model to pay more attention to tail items rather than exclusively optimizing for ground-truth. Overall, our method enables a more nuanced optimization and is mathematically comparable to the direct preference optimization (DPO) in LLMs. Our extensive experiments on three public datasets show our method outperforms fourteen baselines in overall performance and improves the performance on tail items by a large margin. Our code is available at https://anonymous.4open.science/r/R2Rec-0AE0.

## CCS CONCEPTS

• **Information systems** → **Recommender systems**.

## KEYWORDS

Sequential recommendation, tail item, language models and vision models, reweighting and reembedding

**ACM Reference Format:**
Anonymous Author(s). 2025. Reweighting and Reembedding are needed for Tail-item Sequential Recommendation. In *Proceedings of the 2025 ACM Web Conference (WWW'25), April 28 – May 2, 2025, Sydney, Australia.* ACM, New York, NY, USA, 12 pages. https://doi.org/XXXXXXX.XXXXXXX

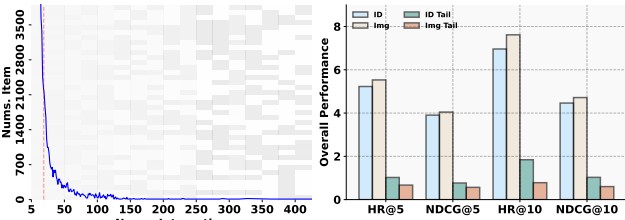

**Figure 1: The left part illustrates the long-tailed distribution of items in the *Amazon* dataset, where the blocks in the background represent the average embedding of items with different interaction counts. It shows 80% items have fewer than 17 interactions (to the left of the red dashed line), and tail item embedding possesses a more uniform-like distribution due to insufficient training. The right part shows the image-based model achieves superior overall performance but performs worse than the ID-based model on tail items.**

## 1 INTRODUCTION

Sequential recommendation aims to capture user preferences from historical interacted sequences to predict the next item. Recently, large language models (LLMs) and large vision models (LVMs) are increasingly applied in sequential recommendation for extracting item features from text or images, given their success in a wide spectrum of applications [5, 6]. Extensive studies have shown the abundant external knowledge encapsulated in the LLMs or LVMs could enhance item representations for better performance. However, those large models suffer more significant performance degradation in tail item recommendation against ID-based methods and this issue largely be overlooked in existing research, hindering their applicability to long-tailed data. As an example, Figure 1 (left) shows the highly skewed and long-tailed item distribution of *Amazon* dataset, where 80% item set have fewer than 17 user interactions and only 5% items have more than 50 interactions. Figure 1 (right) further compares the performance of image-based [1] and ID-based [2] recommendations, showing the image-based model performs poorly on tail items despite its superior overall performance to the ID-based model.

We establish that existing LVMs/LLMs-based models face significant challenges preserving high performance on tail items in sequential recommendation. Firstly, the CE loss focuses exclusively on increasing the likelihood of the target item (ground-truth) while treating all non-ground-truth items as equally incorrect [10, 27], *i.e., all-in ground-truth*. However, user preferences may vary across

---

[1]We use the item image representation generated by CLIP (https://huggingface.co/openai/clip-vit-large-patch14) as the initial item embedding and train a Transformer-based model with the cross-entropy loss for recommendation.
[2]We use item ID as embedding and train a Transformer-based model with the cross-entropy loss for recommendation.

items and inherently require differentiated optimization efforts; moreover, tail items might be sufficiently optimized when compared with popular items; further, some non-ground-truth items might still be preferred by users, provided they are exposed to the users. The insufficient optimization of tail items is reflected in Figure 1 (left), where the average embeddings of popular items (with more interactions) are more distinct and sharper; in contrast, the average embeddings of tail items (with very few interactions) tend to be uniform. Secondly, although the item representations derived from LLMs/LVMs encompass enriched knowledge, such prior knowledge encapsulated in these representations could dominate optimization directions. Since recommendation models rely more on the knowledge of LLMs and LVMs than on the collaborative filtering signals learned from historical records, such external knowledge transfer may adversely impact the models, causing performance degradation on tail items, i.e., *knowledge transfer tax*. Although some efforts [17, 42] apply auxiliary information from other domains to improve tail-item recommendation, the improvement is rather limited for image-based recommendation, leaving the issue underexplored.

To address the above issues, we propose R²REC, a simple yet efficient approach based on reweighting and reembedding to the sequential recommendation. Specifically, we first reinitialize tail item embedding using a standard Gaussian distribution to avoid the negative effect of external knowledge transfer. Then, we incorporate a reweighting function in the vanilla cross-entropy (CE) loss. The function adaptively adjusts score distribution during training, pushing the model towards paying more attention to tail items. As such, it considers both recommendation accuracy and diversity on tail items, enforcing nuanced optimization and alleviating the issue caused by insufficient training on tail items. This function formulation is potentially comparable to the preference alignment algorithms (*e.g.,* RLHF [29], DPO [31]), which are designed to guide the model toward recommending tail items in LLMs.

In a nutshell, we make the following contributions in this paper:

- We first explore the *all-in ground-truth* and *knowledge transfer tax* issues with LLM/LVM-based recommenders. We also provide a mathematical analysis to investigate the performance degradation of LLM/LVM-based recommenders on tail items from those two perspectives.
- We propose a simple yet effective approach, R²REC, which reinitializes tail item embedding and reweights the CE loss adaptively during model training to address the above-identified issues.
- We conducted extensive experiments on three real-world datasets to demonstrate the advantages of our approach in promoting tail item preferences. The results show our proposed method outperforms all the 14 baselines and improves the recommendation performance on tail items by a large margin.

## 2 RELATED WORK

**Sequential Recommendation.** Sequential recommendation aims to recommend the next item the user prefers based on the historical interaction sequences. Early studies focus on the Markov chain method, which models a user behavior sequence as a Markov Decision Process (MDP) for next-item prediction [33, 34]. Given the promising results of deep neural networks in a variety of tasks,

many studies seek to apply GRU, LSTM, Transformer, and their variants to sequential recommendation [8, 14, 35]. Among those techniques, graph neural networks, diffusion methods, and contrastive learning have attracted increasing attention in sequential recommendation research [19, 20, 39, 40]. Recently, natural language processing (NLP) and computer vision (CV) applications have witnessed great successes of large pre-training models. This inspires emerging studies on applying LLMs or LVMs to sequential recommendation [6, 11]. As the examples, UnisRec [11] and Recformer [18] follow a similar procedure: they first pre-train a language model via item text (e.g., title, categories, brands) on source domains and then fine-tune it on target domain data for making commendations. P5 [5] is another paradigm that defines the sequential recommendation as a next-token generation task; it fine-tunes LLMs with prompt engineering for next-item prediction. Based on P5, VIP5 [6] further incorporates image representation as the embedding for sequential recommendation. For image-based or hybrid sequential recommendation (which leverages both text and images), the majority of studies [6, 7, 38] adopt LLMs or LVMs to obtain items' text and image representations. Afterwards, a dedicated cross-attention module is employed for representation fusion. In view of the immense computational cost and memory footprint of LLMs-based or LVMs-based recommenders, MMMLP [22] only applies MLP layers as the backbone for sequential recommendation. Other studies use tailored adapters for parameter-efficient fine-tuning and recommendations [1, 24].

**Tail-Item Sequential Recommendation.** The tail item problem refers to the phenomenon that only very few head items receive vast attention while the majority of items (a.k.a., tail items) are unpopular and attract very limited interactions. The tail item problem widely and chronically exists in online services, incurring popularity basis and impairing recommendation performance. Existing efforts to tackle this problem mainly focus on introducing auxiliary information to enhance item representations, especially tail item presentations, for sequential recommendation. Tyipcal auxiliary information include assistance relationships from head-item [13, 15, 26], similar items [41] or similar sequences [12], and semantic information from LLMs [25], LVMs [2] or knowledge graph (KG) [44]. Despite above efforts, existing recommenders commonly suffer degraded performance on tail items. Moreover, none of them offer a theoretical discussion to investigate the underlying causes or provide a solution to address the issue.

## 3 METHODOLOGY

### 3.1 Problem Statement

**Sequential Recommendation.** Let the user set and item set be $u \in \mathcal{U}$ and $i \in \mathcal{I}$, respectively. Given a chronologically organised sequence of historically interacted items of user $u$, *i.e.,* $s_u = [i_1, i_2, ..., i_\ell]$, sequential recommendation aims to predict the probability that user $u$ will be interested in item $i$ at the next step $\ell + 1$, *i.e.,* $P(i_{\ell+1}|s) = Q_\theta(i_{\ell+1}|s)$, where $Q_\theta(\cdot)$ is a sequential recommendation model parameterized by $\theta$.

The predominant method to estimate $\theta$ is to train model $Q_\theta$ on a vast corpus of historical interaction sequences using maximum likelihood estimation. Given the target item distribution $P$, the objective of model training is to minimize the cross-entropy between

$P$ and $Q_\theta$, as formalized below:

$$\mathcal{L}_{\text{CE}}(P, Q_\theta) = -\mathbb{E}_{i_{\ell+1} \sim P}[\log Q_\theta(i_{\ell+1}|s)] \qquad (1)$$

## 3.2 Deficiency of CE Loss and Text-based or Image-based Embedding

**All-in Ground-truth.** Following the standard CE loss specified in Eq. (1), we derive the gradient cross-entropy with regard to model parameters $\theta$ as follows:

$$\nabla_\theta \mathcal{L}_{\text{CE}}(P, Q_\theta) = -\mathbb{E}_{i_{\ell+1} \sim P}\left[\frac{\nabla_\theta \log Q_\theta(i_{\ell+1}|s)}{\log Q_\theta(i_{\ell+1}|s)}\right] \qquad (2)$$

According to Eq. (2), when minimizing CE loss via gradient descent, $Q_\theta$ is encouraged to assign a high probability to the target item (ground-truth), *i.e.,* high likelihood items under $P$ shall also have high likelihood under $Q_\theta$ [10, 32]. In contrast, the remaining non-ground-truth items will not receive any explicit activation for optimization during the training. It is both undesirable and unreasonable in practice: firstly, the non-interacted items might still be a potential target—it may just be that they haven't been exposed to the user in the past; secondly, not all non-ground-truth items should be considered equal. Ideally, the training should reward potential candidates by increasing their probabilities to varying degrees rather than penalize them by reducing their probabilities to zero. These deficiencies in the CE loss hinder the model from achieving accurate and diverse recommendations [21].

**Knowledge Transfer Tax.** When the training process is sufficient, the external knowledge encapsulated in image/text-based embedding could enhance item representations, improving recommendation performance. For tail items, however, owing to the limited interaction records, the prior knowledge in embeddings may dominate the training process and optimization directions of the model, inducing performance degradation. Specifically, given the historical sequence $s$, let $P(i|s)$ be the predicted probability distribution of item $i$, which can be estimated by the model $Q_\theta(i|s)$, *i.e.*, $P(i|s) = Q_\theta(i|s)$. Suppose $P_c(i|s)$ is the probability distribution of item $i$ predicted based on the knowledge from LLMs or LVMs. We have

$$
\begin{aligned}
P(i|s) &= \frac{P_c(i|s) \cdot \frac{P(i|s)}{P_c(i|s)}}{\sum_{k \in \mathcal{I}} P_c(k|s) \cdot \frac{P(k|s)}{P_c(k|s)}} \\
&= \frac{P_c(i|s) \cdot \frac{P(s|i)}{P_c(s|i)} \cdot \frac{P(i)}{P_c(i)} \cdot \frac{P_c(s)}{P(s)}}{\sum_{k \in \mathcal{I}} P_c(k|s) \cdot \frac{P(s|k)}{P_c(s|k)} \cdot \frac{P(k)}{P_c(k)} \cdot \frac{P_c(s)}{P(s)}} \\
&= \frac{P_c(i|s) \cdot \frac{P(s|i)}{P_c(s|i)} \cdot \frac{P(i)}{P_c(i)}}{\sum_{k \in \mathcal{I}} P_c(k|s) \cdot \frac{P(s|k)}{P_c(s|k)} \cdot \frac{P(k)}{P_c(k)}} = \frac{\zeta_c(s) P(i) P(s|i)}{\sum_{k \in \mathcal{I}} \zeta_c(s) P(k) P(s|k)}
\end{aligned}
\qquad (3)
$$

Deep neural networks typically apply a *softmax* function to the negative log-likelihood to obtain the optimization objective during model training. As such, we could derive the loss function from Eq. (3) and Eq. (1) as follows:

$$
\begin{aligned}
\mathcal{L}_{CE}(P, Q_\theta) &= \mathcal{L}_{CE}(P, P(i|s)) = -\mathbb{E}_{i \sim P}[\log P(i|s)] \\
&= -\mathbb{E}_{i \sim P}\left[\log \frac{\exp(\zeta_c(s) + P(i) + P(s|i))}{\sum_{k \in \mathcal{I}} \exp(\zeta_c(s) + P(k) + P(s|k))}\right]
\end{aligned}
\qquad (4)
$$

where $\zeta_c(s) = \frac{1}{P_c(s)}$. Generally, this term does not significantly impact the training process as the external knowledge from the pre-trained embedding contains limited information relevant to the sequence $s$. However, in tail-item prediction, the values of $P(i)$ and $P(s|i)$ are significantly smaller due to the lower occurrence of tail items in the dataset, causing $\zeta_c(s)$ to dominate the loss function. In this case, the model will rely more on the prior knowledge from LLMs or LVMs instead of the collaborative patterns learned from sequences for recommendation, which, in turn, impairs the recommendation performance on tail items.

## 3.3 R²REC Framework

The framework of our proposed R²REC is depicted in Figure 2, which consists of three main components: (1) the image-based transformer backbone for a sequential recommendation, (2) the reweighting function incorporated with the CE loss and (3) the embedding (incl., reembedding) operation applied on tail items.

**Image-based Recommendation.** In R²REC, we adopt the vanilla Transformer [37] as our backbone. For the prediction layer, we apply a linear projection operation that leverages the updated representation of the sequence's final item, which encapsulates the information of the entire sequence, to predict the next item for recommendation. We obtain the correspondent image of the input item and apply the CLIP [3] image encoder (CLIP-Img) [30] as the embedding layer for item representation initialization. It is formalized as follows:

$$
\begin{aligned}
\mathbf{x}_1, \mathbf{x}_2, ..., \mathbf{x}_\ell &= \text{CLIP-Img}(i_1, i_2, ..., i_\ell) \\
\mathbf{h}_1, \mathbf{h}_2, ..., \mathbf{h}_\ell &= \text{ImgRec}(\mathbf{x}_1, \mathbf{x}_2, ..., \mathbf{x}_\ell) \\
\mathbf{y}_{\ell+1} &= \mathbf{W}\mathbf{h}_\ell^T
\end{aligned}
\qquad (5)
$$

where $[i_1, ..., i_\ell]$ and $[\mathbf{x}_1, ..., \mathbf{x}_\ell]$, $\mathbf{x}_k \in \mathbb{R}^{1 \times d}$ are the item image and the correspondent image-based embedding generated by the CLIP. $[\mathbf{h}_1, ..., \mathbf{h}_\ell]$, $\mathbf{h}_k \in \mathbb{R}^{1 \times d}$ are the updated item representation generated by the transformer. $\mathbf{y}_{\ell+1} \in \mathbb{R}^{N \times 1}$ is the predicted target item scores at step $\ell + 1$, $N$ is the total number of items. $\mathbf{W} \in \mathbb{R}^{N \times d}$ is a learnable parameter matrix. The predicted scores will generally undergo a *softmax* function to obtain the probability and then be fed into the CE loss for model optimization.

**Reembedding Operation.** Following Eq. (4), the external knowledge encapsulated in the tail item embeddings will dominate the model optimization directions and hurt the model performance. Consequently, a straightforward recipe is to reinitialize the text-based or image-based tail item embeddings as a standard Gaussian distribution (*i.e.,* $\mathbf{x} \sim \mathcal{N}(0, 1)$) and train them from scratch. This process can be formalized below,

$$
\mathbf{x}_i = \begin{cases} \text{CLIP-Img}(i) & \text{if } i \notin \mathcal{I}_{TL} \\ \sim \mathcal{N}(0, 1) & \text{otherwise} \end{cases}
\qquad (6)
$$

**Reweighting Function.** As discussed in Section 3.2, the standard CE loss treats all the items as equal. This is inappropriate as the tail items are the minority in the training samples and can not acquire insufficient training. Consequently, we propose an efficient reweighting function that adaptively adjusts the predicted probability, optimizing the training process to pay more attention to the

---

[3]https://huggingface.co/openai/clip-vit-large-patch14

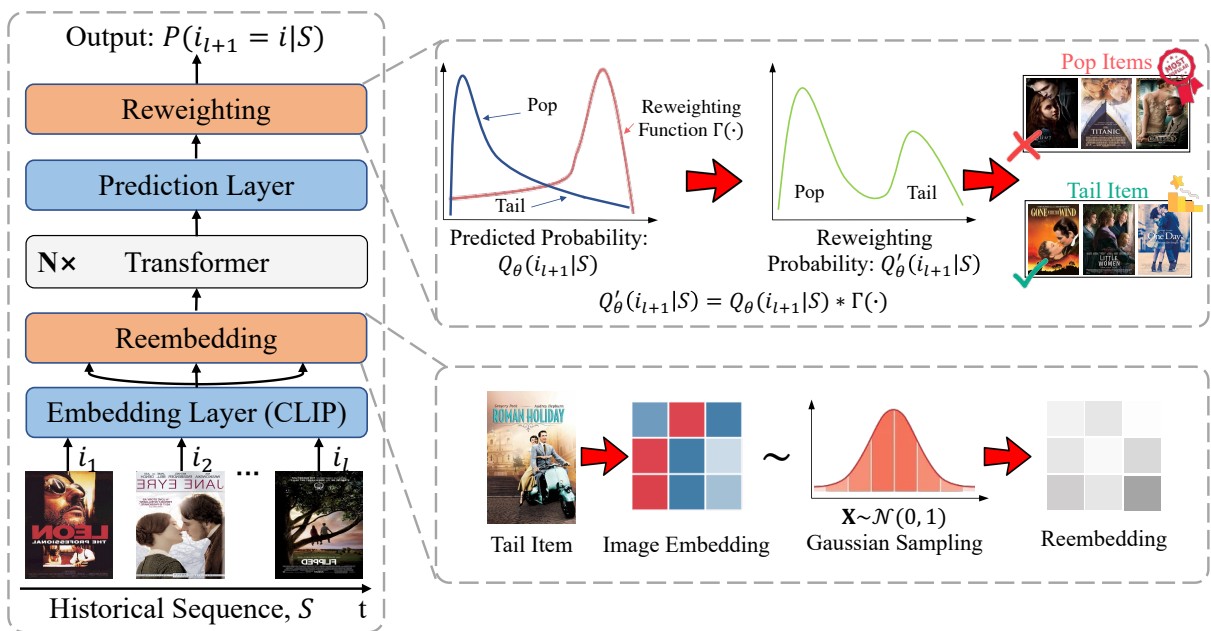

**Figure 2: Framework of $R^2REC$. It adopts Transformer as the backbone (left) and includes two key modules (right),** *i.e.,* **Reembedding and Reweighting.**

tail items. Formally, the CE loss incorporated with the reweighting term can be defined as follows:

$$
\begin{aligned}
\mathcal{L}_{\text{RCE}}(P, Q_\theta) &= -\mathbb{E}_{i_{\ell+1} \sim P}[\log f(Q_\theta(i_{\ell+1}|s))] \\
&= -\mathbb{E}_{i_{\ell+1} \sim P}[\log Q_\theta(i_{\ell+1}|s)\Gamma(i_{\ell+1}, Q_\theta(i_{\ell+1}|s))] \\
&= -\sum_{i \in \mathcal{I}} P(i_{\ell+1}|s)\log Q_\theta(i_{\ell+1}|s)\Gamma(i_{\ell+1}, Q_\theta(i_{\ell+1}|s))
\end{aligned}
$$

$$
s.t \quad \sum_{i \in \mathcal{I}} Q_\theta(i_{\ell+1}|s) = 1 \quad \forall s \in \mathcal{S}; \quad \sum_{i \in \mathcal{I}} \Gamma(i_{\ell+1}, Q_\theta(i_{\ell+1}|s)) = 1
$$

(7)

where $\Gamma(\cdot) \in \mathbb{R}^{N \times N}$ is the reweighting function. Intuitively, the reweighting function is required to (1) be the function of $Q_\theta$; (2) assign larger weight to the tail items and in turn decrease the weight of the head items and (3) the weight distribution should be adjusted dynamically during the model training, *i.e.,* with different weights applied at each epoch. Therefore, we formalize the reweighting function $\Gamma_j(i, Q_\theta(\hat{i}|s))$ at $j$-th training epoch as:

$$
\Gamma_j(i, Q_\theta(\hat{i}|s)) = \frac{\exp(\eta_j(i, Q_\theta(\hat{i}|s))/\tau)}{\sum_{i \in \mathcal{I}} \exp(\eta_j(i, Q_\theta(\hat{i}))/\tau)}
$$

(8)

$$
\eta_j(i, Q_\theta(\hat{i}|s)) = \begin{cases} \eta_{j-1}(i, Q_\theta(\hat{i}|s)) + \alpha_p & \text{if } i \in \{\hat{i}_1^{j-1}, ..., \hat{i}_k^{j-1}\} \text{ and } i \in \mathcal{I}_{TL} \\ \eta_{j-1}(i, Q_\theta(\hat{i}|s)) - \alpha_r & \text{if } i \notin \{\hat{i}_1^{j-1}, ..., \hat{i}_k^{j-1}\} \text{ and } i \in \mathcal{I}_{TL} \\ \alpha_b & \text{others} \end{cases}
$$

(9)

where $i$ is the ground-truth and $\{\hat{i}_1^{j-1}, ..., \hat{i}_k^{j-1}\}$ are the top-$K$ item list at the $(j-1)$-th training epoch predicted by the model $Q_\theta$. We set $K = 5$. $\mathcal{I}_{TL}$ are the tail item set. $\alpha_p, \alpha_r, \alpha_b$ are the plenty, reward, and base factors. We set $\alpha_p = \alpha_b = 1$ and $\alpha_r = 0$. $\tau$ is a temperature

factor to control the shape of the distribution, we set $\tau = 0.5$. We initialize the $\eta_0 = 1$ at the beginning of the training epoch.

**Training Process.** The full training process is summarized in Algorithm 1. Compared to the standard training process, we introduce only two additional operations: reembedding at the first item embedding layer and reweighting at the final loss function calculation stage. Therefore, our method can be easily adapted to a wide range of recommendation models with minimal modifications.

### 3.4 Discussion

We discuss the properties of the proposed reweighting function, providing insights into its advantages over the standard cross-entropy loss, particularly with respect to tail items during optimization.

**More Precise Optimization.** The CE loss is demonstrated to be the ground-truth first, as it treats all non-ground-truth equally, providing no supervised signal for their optimization. To amend this problem, we proposed RCE loss, incorporating a reweighting function to CE loss. Considering the gradient of general RCE formalizes with respect to model parameters $\theta$, we have:

$$
\begin{aligned}
\nabla_\theta \mathcal{L}_{\text{RCE}}(P, Q_\theta) &= -\mathbb{E}_{i_{\ell+1} \sim P}[\log \nabla_\theta Q_\theta(i_{\ell+1}|s)\Gamma(i_{\ell+1}, Q_\theta(i_{\ell+1}|s))] \\
&= -\sum_{i \in \mathcal{I}} \log \nabla_\theta Q_\theta(i_{\ell+1}|s)\mathbb{E}_{i_{\ell+1} \sim P}[\Gamma(i_{\ell+1}, Q_\theta(i_{\ell+1}|s))]
\end{aligned}
$$

(10)

From Eq. (10), we can observe that the model gradients $\nabla_\theta$ will be updated under the supervision of the reweighting function $\Gamma(\cdot)$, which is defined across all the items. Therefore, all the items will participate and contribute to the model optimization process. By designing different $\Gamma(\cdot)$, we can modulate the attention assignment

**Algorithm 1:** $R^2$Rec Traning Process

1: **Input:**
2:     Image-based historical sequence: $[i_1, \cdots, i_\ell]$;
3:     Ground-truth probability: $P_{\ell+1}$;
4:     Tail item set: $\mathcal{I}_{TL}$;
5:     CLIP image encoder: CLIP-Img$(\cdot)$;
6:     Transformer-based backbone ImgRec: $Q_\theta(\cdot)$;
7:     Reweighting function: $\Gamma(\cdot)$;
8:     Learning epochs: $T$;
9:     Optimizer: AdamW$(\cdot)$;
10: **Output:**
11:     Predicted target item: $\hat{i}_{\ell+1}$;
12: **while** $j < T$ **do**
13:     $[\mathbf{x}_1, ..., \mathbf{x}_\ell] = $ CLIP-Img$(i_1, .., i_\ell)$; // Embedding initialization
14:     $\mathbf{x}_j \sim \mathcal{N}(0, 1), i_j \in \mathcal{I}_{TL}$;// Reembedding
15:     $\mathcal{L}_{RCE}(Q_\theta([\mathbf{x}_1, ..., \mathbf{x}_\ell]), P, \Gamma)$;// RCE loss, Eq. (7) to 9
16:     $\theta \leftarrow$ AdamW$(\mathcal{L}_{RCE}, \theta)$; // Parameter update
17:     $j = j + 1$;
18: **end while**

to items. In contrast to the CE loss, RCE endows a more nuanced optimization due to the availability of diverse supervisory weights for all items.

**Connection with Direct Preference Optimization (DPO).** DPO is one of the most popular offline preference optimization methods used for preference alignment in LLMs [31]. Instead of learning an explicit reward model [29], DPO algorithm optimizes the policy in a straightforward manner by reparameterizing the reward function $r(\cdot)$ using a closed-form expression in a supervised manner:

$$r(x, y) = \beta \log \frac{\pi_\theta(y|x)}{\pi_{(ref)}(y|x)} + \beta \log Z(x) \tag{11}$$

where $\pi_\theta(\cdot|x)$ and $\pi_{\text{ref}}(\cdot|x)$ are the policy model and reference model, respectively. $\beta$ is the coefficient of the partition function or the normalizing constant $Z(x)$.

By incorporating the reward function (Eq. (11)) into the Bradley-Terry (BT) ranking objective formula [4],

$$p(y_w \succ y_l | x) = \frac{1}{1 + \exp(r(y_w) - r(y_\ell))} = \sigma(r(x, y_w) - r(x, y_\ell)) \tag{12}$$

We can cancel out the partition function $Z(x)$, resulting in the objective of DPO with reverse KL divergence below:

$$-\mathbb{E}_{(x, y_w, y_\ell) \sim \mathcal{D}} \left[ \log \sigma \left( \beta \log \frac{\pi_\theta(y_w|x)}{\pi_{\text{ref}}(y_w|x)} - \beta \log \frac{\pi_\theta(y_\ell|x)}{\pi_{\text{ref}}(y_\ell|x)} \right) \right] \tag{13}$$

where $\sigma(\cdot)$ is the sigmoid function. $y_w$ and $y_\ell$ are preference pairs consisting of the approved (win) response and refused (lose) response with regard to the input $x$.

Based on the DPO function, we define $\beta = 1$, thus, the Eq. (13) can be expressed below:

$$-\mathbb{E}_{(x, y_w, y_\ell) \sim \mathcal{D}} \left[ \log \sigma \left( \log \frac{\pi_\theta(y_w|x)}{\pi_{\text{ref}}(y_w|x)} - \log \frac{\pi_\theta(y_\ell|x)}{\pi_{\text{ref}}(y_\ell|x)} \right) \right]$$
$$= -\mathbb{E}_{(x, y_w, y_\ell) \sim \mathcal{D}} \left[ \log \sigma \left( \log \frac{1}{\pi_\theta(y_\ell|x)} * \pi_\theta(y_w|x) \right) \right] \tag{14}$$

As the original DPO only models the pairwise preference comparison (i.e., $\pi_\theta(y_w|x)$ and $\pi_\theta(y_\ell|x)$), instead, we consider all the possible outputs and therefore replace $1/\pi_\theta(y_\ell|x)$ as the reweighting function $\Gamma(y, \pi_\theta(y|x))$, which can also control the preference alignment based on the plenty or reward factors. Moreover, we replace the $\sigma$ as *softmax* function. Therefore, the Eq. (14) can be rewritten into

$$-\mathbb{E}_{(x, y) \sim \mathcal{D}} \left[ \log \frac{\pi_\theta(y_i|x) \gamma(y_i, \pi_\theta(y_i|x))}{\sum_{y_j \in \mathcal{Y}} \pi_\theta(y_j|x) \gamma(y_j, \pi_\theta(y_j|x))} \right]$$
$$= -\mathbb{E}_{y \sim P} [\log Q_\theta(y|x) \Gamma(y, \pi_\theta(y|x))] \tag{15}$$

From this perspective, our reweighting function can be recognized as a preference alignment strategy, steering the model to prioritize tail item outputs.

## 4 EXPERIMENTS

We conduct extensive experiments over three real-world datasets against fourteen baselines for performance evaluation. Overall, we aim to answer the following questions:

- **RQ1.** How does the overall performance of $R^2$Rec and its performance on tail items compare with state-of-the-art sequential recommendation methods?
- **RQ2.** How do tailored reweight and reembedding operations applied in $R^2$Rec affect its performance?
- **RQ3.** How do different hyperparameter settings affect the performance of $R^2$Rec on three datasets?
- **RQ4.** How do item embeddings evolve throughout the various phases of model training?

### 4.1 Datasets and Evaluation Metrics

**Datasets.** We select three subcategories, i.e., *Beauty*, *Toys*, and *Sports*, from Amazon datasets[4] [28] for performance evaluation, as they contain enriched item information (e.g., item title, description, and image, and so on) and user information (e.g., user ratings and reviews, and so on). All of these datasets are widely used in sequential recommendation. Following previous works [2, 6, 14], we recognize the user-item ratings as interactions, and the items and users with fewer than five interaction records are removed. For each user, we organize the filtered interactions chronologically based on the timestamp and adopt the leave-one-out strategy for the training dataset, validation dataset, and testing dataset split, i.e., given a sequence $s = [i_1, i_2, ..., i_{\ell+1}]$, we gather the most recent interaction $i_{\ell+1}$ for model testing, the penultimate interaction $i_\ell$ for model validation, and the remains $[i_1, i_2, ..., i_{\ell-1}]$ for model training. We set the maximum sequence length as 10 for all the datasets. Based on the Pareto principle [3] as the criteria, we set the ratio as 20% to curate the tail item.

---
[4]https://nijianmo.github.io/amazon/index.html

Table 1: Overall performance on the three datasets. The best results are highlighted in boldface, and the second-best results are underlined. "−" means the text modeling module is removed from raw models and only uses image information for recommendation. -txt and -img denote item text, and item image information is considered, respectively, for embedding initialization and recommendation. † means cross-domain transfer learning is applied. ▲% means improvement (%) against the best results excluding the $R^2$REC variants. * denotes a significant improvement over the best baseline results (t-test P<.05).

| Dataset | Amazon Beauty | | | | Amazon Toys | | | | Amazon Sports | | | |
|---|---|---|---|---|---|---|---|---|---|---|---|---|
| | HR@5 | HR@10 | NDCG@5 | NDCG@10 | HR@5 | HR@10 | NDCG@5 | NDCG@10 | HR@5 | HR@10 | NDCG@5 | NDCG@10 |
| GRU4Rec | 0.0164 | 0.0283 | 0.0099 | 0.0137 | 0.0097 | 0.0176 | 0.0059 | 0.0084 | 0.0129 | 0.0204 | 0.0086 | 0.0110 |
| Caser | 0.0232 | 0.0394 | 0.0149 | 0.0201 | 0.0202 | 0.0329 | 0.0121 | 0.0162 | 0.0130 | 0.0222 | 0.0079 | 0.0108 |
| SASRec | 0.0327 | 0.0626 | 0.0240 | 0.0323 | 0.0454 | 0.0655 | 0.0301 | 0.0375 | 0.0172 | 0.0325 | 0.0089 | 0.0138 |
| $S^3$-Rec | 0.0387 | 0.0647 | 0.0244 | 0.0327 | 0.0443 | 0.0700 | 0.0294 | 0.0376 | 0.0251 | 0.0385 | 0.0161 | 0.0204 |
| UniSRec | 0.0476 | 0.0734 | 0.0263 | 0.0331 | 0.0455 | 0.0713 | 0.0254 | 0.0337 | 0.0264 | 0.0457 | 0.0143 | 0.0220 |
| P5 | 0.0494 | 0.0690 | 0.0394 | 0.0412 | 0.0619 | 0.0716 | 0.0312 | 0.0425 | 0.0290 | 0.0381 | 0.0168 | 0.0215 |
| MM-Rec⁻ | 0.0377 | 0.0546 | 0.0224 | 0.0279 | 0.0596 | 0.0779 | 0.0336 | 0.0405 | 0.0279 | 0.0404 | 0.0162 | 0.0201 |
| MMMLP⁻ | 0.0313 | 0.0494 | 0.0211 | 0.0269 | 0.0215 | 0.0338 | 0.0147 | 0.0187 | 0.0157 | 0.0267 | 0.0100 | 0.0136 |
| MMSBR | 0.0331 | 0.0557 | 0.0201 | 0.0273 | 0.0299 | 0.0476 | 0.0183 | 0.0240 | 0.0182 | 0.0318 | 0.0116 | 0.0160 |
| MMMLP | 0.0526 | 0.0754 | 0.0382 | 0.0448 | 0.0588 | 0.0812 | 0.0436 | 0.0488 | 0.0320 | 0.0448 | 0.0219 | 0.0263 |
| MM-Rec | 0.0381 | 0.0552 | 0.0227 | 0.0282 | 0.0603 | 0.0783 | 0.0338 | 0.0414 | 0.0294 | 0.0419 | 0.0174 | 0.0215 |
| MELT | 0.0221 | 0.0428 | 0.0118 | 0.0184 | 0.0284 | 0.0491 | 0.0144 | 0.0211 | 0.0170 | 0.0289 | 0.0103 | 0.0141 |
| CITIES | 0.0487 | 0.0695 | 0.0355 | 0.0422 | 0.0570 | 0.0751 | 0.0426 | 0.0484 | 0.0278 | 0.0417 | 0.0190 | 0.0235 |
| MAN | 0.0535 | 0.0715 | 0.0398 | 0.0456 | 0.0606 | 0.0769 | 0.0449 | 0.0502 | 0.0311 | 0.0430 | 0.0223 | 0.0262 |
| $R^2$REC-txt | 0.0538 | 0.0739 | 0.0398 | 0.0462 | 0.0622 | 0.0807 | 0.0464 | 0.0524 | 0.0287 | 0.0384 | 0.0204 | 0.0235 |
| $R^2$REC-img | 0.0545 | 0.0801 | 0.0402 | 0.0489 | 0.0637 | 0.0867 | 0.0469 | 0.0539 | 0.0315 | 0.0479 | 0.0225 | 0.0276 |
| $R^2$REC-img† | 0.0559* | 0.0806* | 0.0412* | 0.0490* | 0.0641* | 0.0875* | 0.0474* | 0.0559* | 0.0327* | 0.0491* | 0.0229* | 0.0287* |
| ▲% | 4.49% | 6.90% | 3.52% | 7.46% | 5.78% | 7.76% | 5.57% | 11.35% | 2.19% | 9.60% | 2.69% | 9.13% |

**Evaluation Metrics.** We apply two popular recommendation metrics, HR@N (Hit Rate) and NDCG@N (Normalized Discounted Cumulative Gain), for performance evaluation (ref. Appendix A.2 for details). The HR@N measures how many hits are present within the top-N recommended list, which reveals the capability of models in recall. NDCG@N further evaluates the model ranking performance by considering the ranking position of these hits in the list. We set N = 5 and N = 10 to compare the experimental results of our $R^2$REC with the baseline models. Due to the negative sampling evaluation will incur a distinct gap against the practical scenario when the size of negative samples is small [16], we take all the items as candidates for performance comparison.

## 4.2 Baselines and Implementation Details

**Baselines.** We select three categories of methods, including (1) convention ID-based models; (2) image-based models, text-based models, and image-text-based models; and (3) tail-item-oriented methods, for a comprehensive performance evaluation. These methods are highly relevant to our research.

- **ID-based Models.** We select **GRU4Rec**[5] [8], **Caser**[6] [36], **SAS-Rec**[7] [14], **$S^3$-Rec**[8] [45], the four representative ID-based methods for performance evaluation, covering three mainstream neural network architectures, i.e., GRU, CNN, and Transformer.

- **Text-based and Image-based Models.** We include two text-based models (i.e., **UniSRec**[9] [11] and **P5**[10] [5]), two image-based models (i.e., **MM-Rec**$^{-}$[11] [38] and **MMMLP**$^{-}$[12] [22]) and three multi-modality (image and text) models (i.e., **MM-Rec**, **MMSBR**[13] [43], and **MMMLP**). All of them are committed to obtaining the item image or text representation via vision and language models and then dedicate representation fusion modules (e.g., cross-attention) for recommendation.

- **Tail Item Oriented Models.** We compare our $R^2$REC against three tail item sequential recommendation solutions, i.e., **CITES** [13], **MELT**[14] [15], and **MAN** [23]. To tackle the challenge of tail item recommendation, all of them endeavor to introduce more auxiliary information from e.g., head items, context items, and cross-domains, for representation enhancement.

**Implementation Details.** We apply the AdamW optimizer with the learning rate of $5e − 4$ and adopt the warm-up strategy with a step ratio of 0.1. We set the multi-head numbers as 16 and the block numbers as 1. Furthermore, we set the hidden size of Transformer blocks as 768, the same as the embedding size. The dropout ratio is 0.8, and the batch size is 128. As for the hyperparameter $\tau$ in the reweighting function, we set $\tau = 0.5$ and further analyze the efficiency within the scope of 0.1 to 1. We compare text-based[15],

---

[5]https://github.com/hidasib/GRU4Rec
[6]https://github.com/graytowne/caser
[7]https://github.com/kang205/SASRec
[8]https://github.com/RUCAIBox/CIKM2020-S3Rec

[9]https://github.com/RUCAIBox/UniSRec
[10]https://github.com/jeykigung/P5
[11]"-" means the text modeling module is removed from raw models.
[12]https://github.com/Applied-Machine-Learning-Lab/MMMLP
[13]https://github.com/Zhang-xiaokun/MMSBR
[14]https://github.com/rlqja1107/MELT
[15]We extract item titles with maximum tokens of 10 as the corresponding text information and apply CLIP text encoder for item embedding initialization.

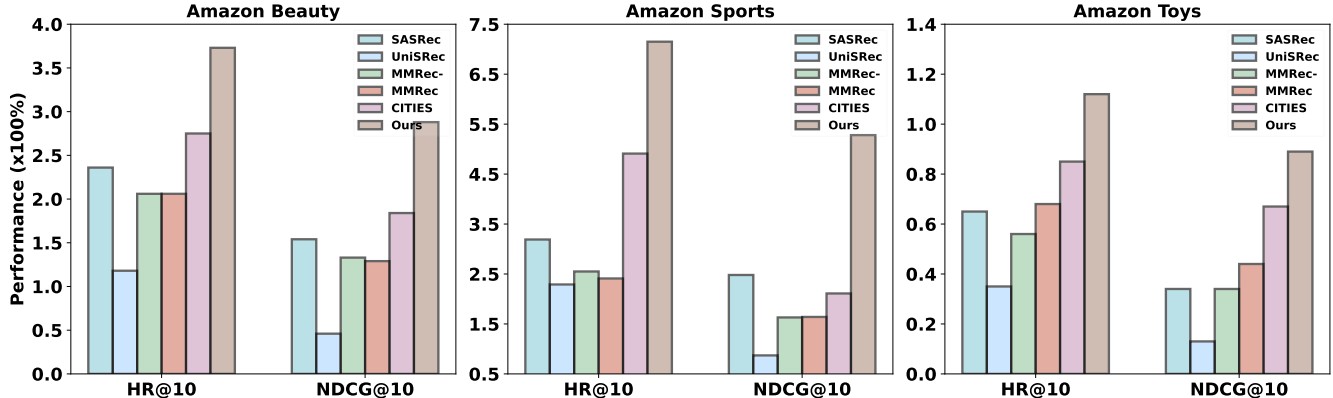

Figure 3: Tail item performance. $R^2$REC superiors to other baselines by a large margin.

image-based, and image-based cross-domain transfer learning[16], three $R^2$REC variants to fully verify the effectiveness of our proposed method.

## 4.3 Overall Performance (RQ1)

**Overall Performance.** The overall comparison results of our $R^2$REC against other baselines are shown in Table 1. Based on the results, we could make the following observations: (1) ID-based solutions can also acquire competitive results; (2) text and image information can further improve the model performance; (3) tail-item-centered models can acquire comparable performance across all the solutions; (4) Our $R^2$REC and its variant, *i.e.,* $R^2$REC with cross-domain transfer learning (ref. Appendix A.1 for the theoretical analysis), achieve best results on all the three datasets, illustrate the effectiveness of the proposed method.

**Tail Item Performance.** Figure 3 depicts the tail item performance on three datasets (Due to the space limitation, the full results are shown in Appendix C.1, Table 4). Compared to the ID-based model, the tail item performance shows varying degrees of decline when incorporating text and image representation extracted by LLMs and LVMs. Our method outperforms tail-item-based solutions by a large margin, fully demonstrating the superiority of reweighting and reembedding strategies.

## 4.4 Ablation Studies (RQ2)

To fully demonstrate the effectiveness of reweighting and reembedding operations, we remove them from $R^2$REC incrementally and also incorporate them into other baselines, observing the performance variation.

- **w/o Reweight.** Removing reweighting operation from $R^2$REC.
- **w/o Reembed.** Removing reembedding operation from $R^2$REC.
- **w R2.** Considering both reweighting and reembedding operations on other baselines[17].

Table 2: Item performance on the *Amazon Beauty* datasets. The best results are highlighted in boldface, and the second-best results are underlined. † means cross-domain transfer learning is applied.

| Dataset | Amazon Beauty | | | |
| | All | | Tail | |
| | HR@10 | NDCG@10 | HR@10 | NDCG@10 |
|---|---|---|---|---|
| $R^2$REC† | **0.0806** | **0.0490** | **0.0399** | 0.0307 |
| $R^2$REC | 0.0801 | 0.0489 | 0.0373 | 0.0288 |
| w/o reweight | 0.0765 | 0.0475 | 0.0194 | 0.0129 |
| w/o reembed | 0.0728 | 0.0443 | 0.0302 | 0.0192 |
| w/o R2 | 0.0762 | 0.0472 | 0.0078 | 0.0060 |
| SASRec w. R2 | 0.0693 | 0.0445 | 0.0371 | 0.0299 |
| MM-Rec⁻ w. R2 | 0.0589 | 0.0411 | 0.0291 | 0.0183 |
| MMMLP⁻ w. R2 | 0.0516 | 0.0386 | 0.0255 | 0.0194 |
| MM-Rec w. R2 | 0.0549 | 0.0364 | 0.0310 | 0.0213 |
| MMMLP w. R2 | 0.0764 | 0.0457 | 0.0336 | 0.0233 |
| CITIES w. R2 | 0.0663 | 0.0401 | 0.0189 | 0.0254 |
| MAN w. R2 | 0.0724 | 0.0484 | 0.0388 | **0.0315** |

Table 2 shows the ablation results[18]. Note that removing either reembedding or reweighting operations leads to a measurable decline in the performance of all items, with a particularly notable impact on tail items. As the reweighting strategy is dedicated to facilitating the optimization of tail items, removing the reweighting function from $R^2$REC results in a significant decline in the performance of tail items compared to the overall performance. Besides, from Table 1 and Table 2, we could find that these two operations also improve the other baselines' tail item performance, though the overall performance improvement is limited.

---

[16]We first pre-train the backbone on *Amazon Pantry*, *Amazon Clothing*, and *Amazon Magazine* datasets. Then, fine-tune it on the target domain.
[17]Since some baselines (*e.g.,* UniSRec, MELT, etc) do not utilize CE loss for model optimization, we exclude them from this experiment.

[18]As page limitation, please ref. Appendix C.2 Table 5 for the results on *Amazon Toys* and *Amazon Sports* datasets, where we could obtain the same observations and conclusions.

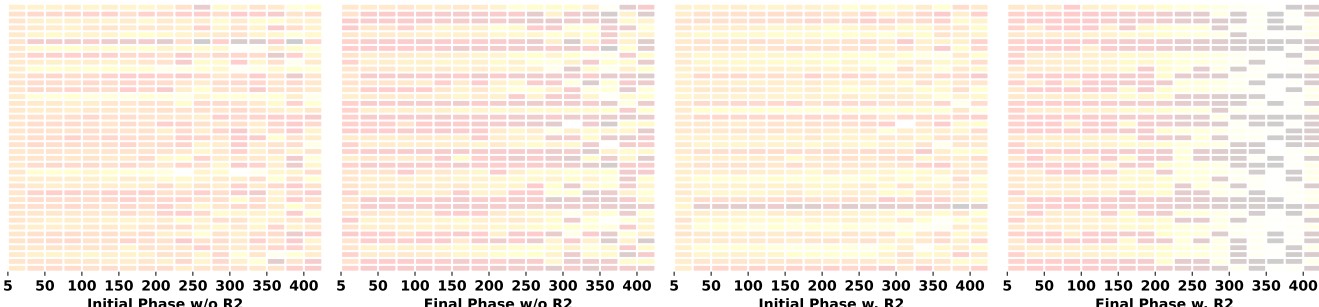

| 5 | 50 | 100 | 150 | 200 | 250 | 300 | 350 | 400 | 5 | 50 | 100 | 150 | 200 | 250 | 300 | 350 | 400 | 5 | 50 | 100 | 150 | 200 | 250 | 300 | 350 | 400 | 5 | 50 | 100 | 150 | 200 | 250 | 300 | 350 | 400 |
|---|---|---|---|---|---|---|---|---|---|---|---|---|---|---|---|---|---|---|---|---|---|---|---|---|---|---|---|---|---|---|---|---|---|---|---|

Initial Phase w/o R2    Final Phase w/o R2    Initial Phase w. R2    Final Phase w. R2

**Figure 4: Visualization of item embedding with different interaction counts (from 5 to 400) at the initial and final stages of model training. The model w. R2 (*i.e.,* with reembedding and reweighting strategies) acquires a more sharp distribution on tail item (the item with fewer interactions) embeddings against the model w/o R2 (*i.e.,* without reembedding and reweighting operations) after training.**

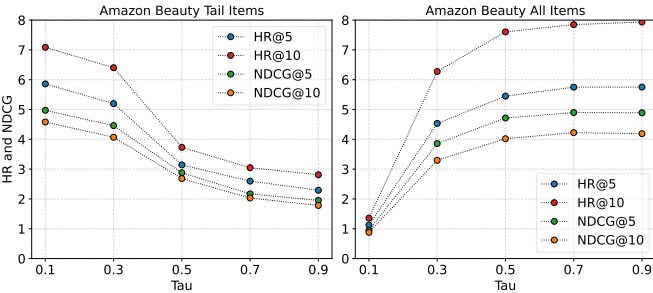

**Figure 5: The performance with varying $\tau$ on Amazon *Beauty* dataset. A moderate $\tau$ achieves the right balance between overall performance and tail item performance.**

## 4.5 Impact of Hyperparameter (RQ3)

We investigate the impact of hyperparameter $\tau$ in Eq. (9) on the final performance (ref. Appendix C.3 Figure 6 for the results on all three datasets). Mathematically, $\tau$ controls the shape of weight distribution and further decides the attention assignment to the samples. As the hyperparameter $\tau$ decreases the weight distribution converges toward a point mass, thereby allocating more attention to the tail items. Conversely, a larger value of $\tau$ will lead to a more uniform weight distribution, ensuring that each item will be treated equivalently. Due to insufficient training on tail items in comparison to popular items, the performance of tail items suffers from significant performance degeneration. Observing Figure 5, we could draw the consistent conclusion that as $\tau$ decreases the tail items' performance can be improved while the overall performance will undergo a substantial decline. Overall, there is a trade-off between the performance on tail items and overall performance, a moderate $\tau$ (such as 0.5) is recommended to maintain a balance between them.

## 4.6 Embedding Visualization (RQ4)

To further explore the merits of reembedding and reweighting methods in tail item sequential recommendation, we examine the impact of removing or incorporating these operations on item embedding evolution. Specifically, we present the average item embeddings from the *Amazon Beauty* dataset, spanning interaction counts from 5 (*i.e.,* tail items) to 400 (head items), observing the embedding variation a during the model training process, as shown in Figure 4. We can observe that in the initial phase of the model training, the item embeddings exhibit a uniform distribution for both models with or without reembedding and reweighting operations. As training progresses, the head items (items with numerous interactions) and tail items (items with fewer interactions) in our $R^2\text{Rec}$ are optimized sufficiently, resulting in a more sharply concentrated embedding distribution. In contrast, for the model without remembedding and reweighting operations, the tail item embeddings demonstrate minimal variation after training. Consequently, we argue that the proposed two operations can enhance the model learning ability in tail items.

## 5 CONCLUSION

This work attempts to alleviate the tail item performance degeneration on image-based sequential recommendations. To instantiate this idea, we first analyze the deficiency of standard CE loss and image-based or text-based embeddings, respectively, then propose *all in ground-truth* and *knowledge transfer tax* two perspectives contributing to this problem. From these two considerations, we further propose reweighting and reembedding functions for recommendation, named $R^2\text{Rec}$. Specifically, instead of treating the head items and tail items equally and focusing solely on the ground-truth, reweighting strategy allows the model to adaptively assign more attention to tail items during the model optimization, alleviating the insufficient training of tail items. Reembedding operation initializes the tail item embedding via a standard Gaussian distribution, tackling the negative transfer of external knowledge encapsulated in LLMs and LVMs. Theoretically, our reweighting function is similar to DPO in LLMs preference alignment but could achieve a more precise optimization. Comprehensive experiments on three public datasets manifest that our $R^2\text{Rec}$ is superior to the baselines on overall performance and tail items. Furthermore, when integrated with other baselines, our method can achieve additional performance improvements.

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

# A PRELIMINARIES

## A.1 Cross Domain Transfer Learning

**Pre-trained knowledge from different domains could facilitate the model performance via transfer learning.** Given a sequence $s$, the target item of prediction is $i$, we define the probability that the model can make the correct prediction based on the knowledge from $k$-th source domain is $P(f_k(i|s)) = 1 - \epsilon$, where $\epsilon$ is the probability of errors. Assuming each source domain is independent and the error probabilities are the same, then we have,

$$P(H(n) \leq m) = \sum_{i=0}^{m} C_n^i (1 - \epsilon)^i \epsilon^{n-i} \tag{16}$$

where $H(n)$ is the number of source domains that can share auxiliary knowledge to assist the model in making the correct prediction. $n$ is the total number of source domains. Based on the Hoeffding's inequality [9], we have,

$$P(H(n) \leq (p - \delta)n) \leq \exp(-2\delta^2 n) \tag{17}$$

Let $\delta = \frac{(1-2\epsilon)}{2}, m = \frac{n}{2}$, we substitute Equation 17 with Equation 16,

$$P(H(n) \leq n/2) \leq \exp(-\frac{1}{2}n(1 - 2\epsilon)^2) \tag{18}$$

Note that as the number of source domains (*i.e., n*) increases, the probability that more than half of the domains contribute useful information to aid the model in making accurate predictions increases. Therefore, we believe the transferred knowledge from cross-domains can enhance the model performance in downstream tasks.

## A.2 Evaluation Metrics for Recommendation

To compare the performance of each baseline to our $R^2$REC, we select Hit Rate (HR@$K$) and Normalized Discounted Cumulative Gain (NDCG@$K$), the most popular evaluation metrics in recommender system, which reveal the recommendation precision and ranking quality of models. The larger the values of these metrics, the better the performance is. Let $R_K$ be the recommended list including the items with Top $K$ predicted scores and $R$ be the ground-truth, *i.e.,* the item the user clicked in the next step. Suppose we have $N$ samples, HR and NDCG can be calculated as follows:

$$\text{HR@}K = \frac{\sum_{i=1}^{N} |R_{K_i} \cap R_i|}{N}$$

$$\text{NDCG@}K = \frac{1}{N} \sum_{i=1}^{N} \frac{1}{Z} \text{DCG@}K = \frac{1}{N} \sum_{i=1}^{N} \left( \frac{1}{Z} \sum_{j=1}^{K} \frac{2^{r_{ij}} - 1}{\log_2(j + 1)} \right) \tag{19}$$

where $r_{ij} \in \{0, 1\}$ indicates whether the ground-truth number $j$ appears in the recommended list for sample $i$. $Z$ is a normalization

constant that is the maximum possible value of DCG. The value of NDCG@$K$ is set to 0 when the rank exceeds $K$.

# B NOTATIONS

We summarize the main notations used in this paper in Table 3.

**Table 3: Major notation.**

| | |
|---|---|
| $i$ | Item |
| $u$ | User |
| $s$ | User interacted historical sequence |
| $\mathcal{I}$ | Item set |
| $\mathcal{U}$ | User set |
| $\mathcal{S}$ | Sequence set |
| $ID$ | Item ID |
| $txt$ | Item text information *e.g.,* title, description and so on |
| $img$ | Item image information |
| $\mathbf{x}_i$ | $i$-th item embedding |
| $\mathbf{h}_i$ | $i$-th item representation |
| $\mathbf{y}$ | Prediction score of models |
| $P$ | The ground-truth probability distribution |
| $Q_\theta$ | The model predicted probability distribution |
| $\Gamma(\cdot)$ | Reweightning function |

# C EXPERIMENTS

## C.1 Tail Item Recommendation Performance

Table 4 presents the tail item performance on three datasets, we could find that our $R^2$REC and its variants achieve the best results against other baselines. As the representative solutions of tail-item sequential recommendation, MAN and CITIES acquire competitive performance compared with other methods. UniSRec shows the poorest performance across all baselines, as the raw text contains excessive noise compared to item images, which severely hampers its effectiveness on tail items.

## C.2 Ablation Studies

Table 5 presents the ablation studies on three datasets. We could observe that removing any dedicated module results in a drop in both overall and tail item performance. Conversely, integrating our reembedding and reweighting operations into other baselines significantly enhances tail item performance, effectively showcasing the effectiveness of our proposed method.

## C.3 Impact of Hyperparameter

The model performance across varying $\tau$ on three datasets is displayed in Figure 6. There is a trade-off between overall performance and tail item performance, thus, a moderate value (*e.g.,* 0.5) is recommended to balance both considerations.

**Table 4: Tail item performance on the three datasets. The best results are highlighted in boldface, and the second-best results are underlined. "−" means the text modeling module is removed from raw models and only uses image information for recommendation. -id, -txt, and -img denote item id, item text, and item image information are used respectively for embedding initialization and recommendation. † means cross-domain transfer learning are applied.**

| Dataset | Amazon Beauty | | | | Amazon Toys | | | | Amazon Sports | | | |
|---|---|---|---|---|---|---|---|---|---|---|---|---|
| | HR@5 | HR@10 | NDCG@5 | NDCG@10 | HR@5 | HR@10 | NDCG@5 | NDCG@10 | HR@5 | HR@10 | NDCG@5 | NDCG@10 |
| SASRec | 0.0165 | 0.0236 | 0.0118 | 0.0154 | 0.0289 | 0.0319 | 0.0238 | 0.0248 | 0.0045 | 0.0065 | 0.0027 | 0.0034 |
| UniSRec | 0.0045 | 0.0118 | 0.0022 | 0.0046 | 0.0074 | 0.0229 | 0.0037 | 0.0087 | 0.0014 | 0.0035 | 0.0006 | 0.0013 |
| MM-Rec⁻ | 0.0157 | 0.0206 | 0.0118 | 0.0133 | 0.0191 | 0.0255 | 0.0142 | 0.0163 | 0.0044 | 0.0056 | 0.0030 | 0.0034 |
| MMMLP⁻ | 0.0102 | 0.0184 | 0.0077 | 0.0103 | 0.0133 | 0.0166 | 0.0115 | 0.0126 | 0.0017 | 0.0017 | 0.0012 | 0.0012 |
| MMMLP | 0.0137 | 0.0182 | 0.0105 | 0.0112 | 0.0152 | 0.0191 | 0.0108 | 0.0120 | 0.0022 | 0.0056 | 0.0022 | 0.0025 |
| MM-Rec | 0.0161 | 0.0206 | 0.0115 | 0.0129 | 0.0201 | 0.0241 | 0.0151 | 0.0164 | 0.0056 | 0.0068 | 0.0040 | 0.0044 |
| MELT | 0.0130 | 0.0203 | 0.0086 | 0.0111 | 0.0199 | 0.0259 | 0.0113 | 0.0132 | 0.0032 | 0.0048 | 0.0024 | 0.0029 |
| CITIES | 0.0221 | 0.0275 | 0.0119 | 0.0184 | 0.0284 | 0.0491 | 0.0144 | 0.0211 | 0.0072 | 0.0085 | 0.0063 | 0.0067 |
| MAN | 0.0175 | 0.0210 | 0.0135 | 0.0146 | 0.0232 | 0.0284 | 0.0189 | 0.0206 | 0.0063 | 0.0078 | 0.0048 | 0.0060 |
| $R^2$Rec-id | **0.0357** | 0.0371 | **0.0294** | 0.0299 | 0.0534 | 0.0568 | 0.0460 | 0.0471 | 0.0073 | 0.0122 | 0.0058 | 0.0074 |
| $R^2$Rec-txt | 0.0309 | 0.0352 | 0.0263 | 0.0276 | 0.0581 | 0.0687 | 0.0471 | 0.0505 | 0.0073 | 0.0083 | 0.0056 | 0.0059 |
| $R^2$Rec-img | 0.0314 | 0.0373 | 0.0268 | 0.0288 | **0.0604** | **0.0715** | **0.0491** | **0.0528** | 0.0112 | 0.0138 | 0.0080 | 0.0089 |
| $R^2$Rec-img† | 0.0340 | **0.0399** | 0.0288 | **0.0307** | 0.0601 | 0.0696 | 0.0488 | 0.0518 | **0.0135** | **0.0141** | **0.0111** | **0.0112** |

**Table 5: Item performance on the three datasets. The best results are highlighted in boldface, and the second-best results are underlined. "−" means the text modeling module is removed from raw models and only uses image information for recommendation. "w/o" and "w." mean with and without the specific module. † means cross-domain transfer learning are applied.**

| Dataset | Amazon Beauty | | | | Amazon Toys | | | | Amazon Sports | | | |
|---|---|---|---|---|---|---|---|---|---|---|---|---|
| | All | | Tail | | All | | Tail | | All | | Tail | |
| | HR@10 | NDCG@10 | HR@10 | NDCG@10 | HR@10 | NDCG@10 | HR@10 | NDCG@10 | HR@10 | NDCG@10 | HR@10 | NDCG@10 |
| $R^2$Rec† | **0.0806** | **0.0490** | **0.0399** | **0.0307** | **0.0875** | **0.0559** | 0.0696 | 0.0518 | **0.0491** | **0.0287** | **0.0141** | **0.0112** |
| $R^2$Rec | 0.0801 | 0.0489 | 0.0373 | 0.0288 | 0.0867 | 0.0539 | **0.0715** | **0.0528** | 0.0479 | 0.0276 | 0.0138 | 0.0089 |
| w/o reweight | 0.0765 | 0.0475 | 0.0194 | 0.0129 | 0.0853 | 0.0541 | 0.0393 | 0.0251 | 0.0486 | 0.0286 | 0.0020 | 0.0012 |
| w/o reembed | 0.0728 | 0.0443 | 0.0302 | 0.0192 | 0.0835 | 0.0524 | 0.0568 | 0.0471 | 0.0435 | 0.0251 | 0.0075 | 0.0048 |
| w/o R2 | 0.0762 | 0.0472 | 0.0078 | 0.0060 | 0.0857 | 0.0538 | 0.0108 | 0.0061 | 0.0475 | 0.0276 | 0.0003 | 0.0003 |
| SASRec w. R2 | 0.0693 | 0.0445 | 0.0371 | 0.0299 | 0.0733 | 0.0489 | 0.0568 | 0.0471 | 0.0240 | 0.0157 | 0.0128 | 0.0107 |
| MM-Rec⁻ w. R2 | 0.0589 | 0.0411 | 0.0291 | 0.0183 | 0.0779 | 0.0452 | 0.0574 | 0.0439 | 0.0373 | 0.0197 | 0.0063 | 0.0059 |
| MMMLP⁻ w. R2 | 0.0516 | 0.0386 | 0.0255 | 0.0194 | 0.0530 | 0.0491 | 0.0403 | 0.0326 | 0.0317 | 0.0221 | 0.0043 | 0.0029 |
| MM-Rec w. R2 | 0.0549 | 0.0364 | 0.0310 | 0.0213 | 0.0806 | 0.0478 | 0.0595 | 0.0457 | 0.0415 | 0.0269 | 0.0087 | 0.0083 |
| MMMLP w. R2 | 0.0764 | 0.0457 | 0.0336 | 0.0233 | 0.0831 | 0.0539 | 0.0559 | 0.0481 | 0.0461 | 0.0267 | 0.0089 | 0.0068 |
| CITIES w. R2 | 0.0663 | 0.0401 | 0.0189 | 0.0254 | 0.0718 | 0.0457 | 0.0613 | 0.0512 | 0.0368 | 0.0210 | 0.0121 | 0.0084 |
| MAN w. R2 | 0.0724 | 0.0484 | 0.0388 | **0.0315** | 0.0730 | 0.0488 | 0.0574 | 0.0471 | 0.0459 | 0.0251 | 0.0095 | 0.0079 |

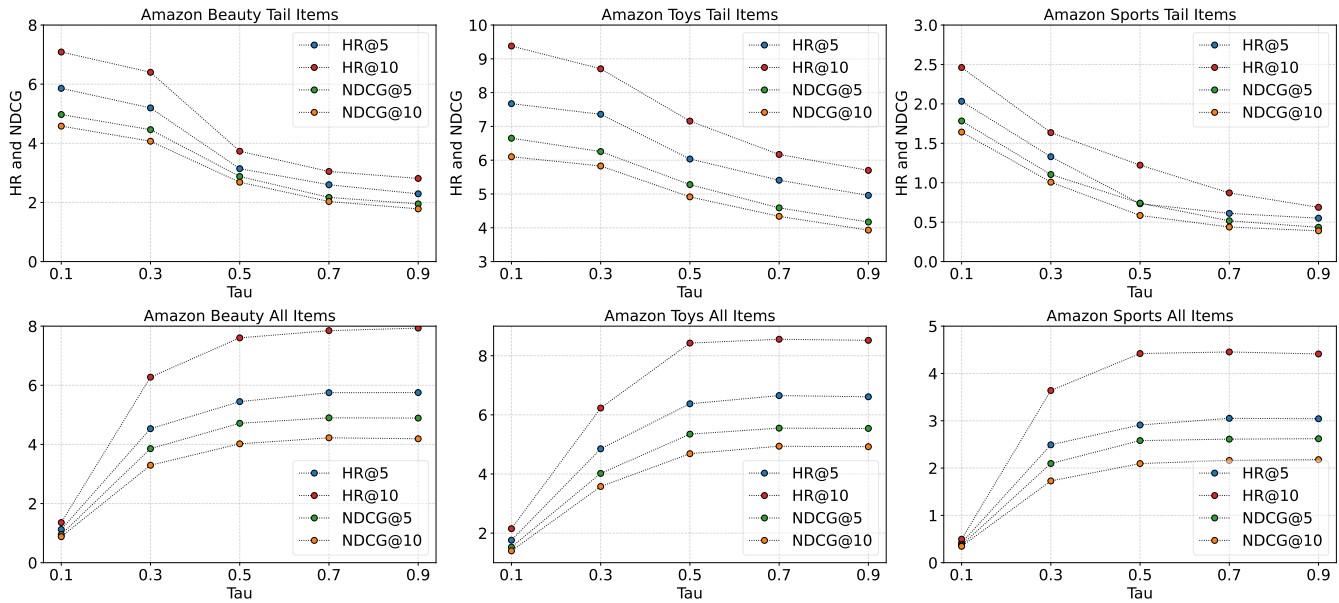

**Figure 6: The performance with varying $\tau$.**

