# OpenReview forum: "Reembedding and Reweighting are Needed for Tail Item Sequential Recommendation"
_ACM.org/TheWebConf/2025/Conference — WWW 2025 Poster_

### Official Review · Reviewer_pgQT · 2024-11-29

**Novelty:** 4
**Technical Quality:** 4

**Review:**

This paper identifies performance challenges for sequential recommendation, particularly with tail items, due to biases in standard CE loss and dominance of pre-trained knowledge. To address this, this paper proposes a method combining reembedding of tail items with Gaussian initialization and a reweighting function to better optimize tail items during training. Extensive experiments on three datasets demonstrate that the proposed method significantly outperforms fourteen baselines, especially on tail item performance.

Strengthens:
1. The paper is well-organized and written in a clear and fluent manner.
2. The experimental comparisons are comprehensive, effectively validating the proposed method's effectiveness.

Weaknesses:
1. It lacks a comparison with LLM-based recommendation methods.
2. The explanation of the transfer tax issue needs to be more explicit.

**Questions:**

1. In Equation (3), there seems to be an issue. Under the condition $P(k|s) = Q_{\theta}(k|s)$, how is it ensured that $\sum_{k \in \mathcal{I}} P(k|s) = 1$ holds before applying the softmax function?
2. Since $\zeta_c(s) = \frac{1}{P_c(s)}$ and $P_c(s)$ is still related to the user's historical interaction sequence, why is it stated that the prior knowledge from LLMs or LVMs conflicts with the collaborative interaction patterns learned by the recommender system, e.g., the former dominates the loss function, leading to degraded performance?
3. For tail items, it is precisely the lack of prior knowledge about interaction patterns that necessitates injecting additional knowledge. The reembedding operation transforms these embeddings into random embeddings following a Gaussian distribution, which results in losing the knowledge injected by the LLM. This seems highly counterintuitive. Could you provide a more detailed explanation of this step?
4. The insight behind reweighting is to enhance the gradient weights of tail items by identifying them. Recent studies have explored improving the performance of tail categories through gradient-driven methods, such as [1][2][3]. Therefore, it is necessary to conduct a deep discussion of these relevant works.
5. The paper is framed from the perspective of LLMs and LVMs, but the experimental comparison lacks comparisons with LLM-based or LVM-based recommendation methods. Furthermore, experiments should also be conducted to explore replacing the CLIP model in the architecture with other LLMs or LVMs.
6. From the hyperparameter $\tau$ experiments, the performance trend of R2REC on tail items appears to be opposite to that on all items. Given the architecture of R2REC, does the performance improvement for tail items come at the expense of head items?

[1] Tan J, Li B, Lu X, et al. The equalization losses: Gradient-driven training for long-tailed object recognition[J]. IEEE Transactions on Pattern Analysis and Machine Intelligence, 2023.

[2] Li B, Yao Y, Tan J, et al. Equalized focal loss for dense long-tailed object detection[C]//Proceedings of the IEEE/CVF conference on computer vision and pattern recognition. 2022: 6990-6999.

[3] Tan J, Lu X, Zhang G, et al. Equalization loss v2: A new gradient balance approach for long-tailed object detection[C]//Proceedings of the IEEE/CVF conference on computer vision and pattern recognition. 2021: 1685-1694.

**Reviewer Confidence:**

3: The reviewer is confident but not certain that the evaluation is correct

**Scope:**

4: The work is relevant to the Web and to the track, and is of broad interest to the community

---

### Official Review · Reviewer_WLLT · 2024-12-01

**Novelty:** 3
**Technical Quality:** 3

**Review:**

The paper presents R2Rec, a framework designed to enhance sequential recommendation systems, particularly for tail items. The framework integrates three primary components: a Transformer Backbone, a Reweighting Function, and a Reembedding Operation. While the reembedding method contributes to the overall approach, it could be considered relatively simplistic and lacks novelty compared to other recent techniques in the field. Additionally, the experimental analysis could be more comprehensive. Specifically, the paper could benefit from providing insights into the proportion of long-tail items in the final recommendation results.

**Questions:**

1. The explanation of Equation (5) is somewhat unclear, making it difficult to fully grasp the mechanics of the proposed method.
2. While the model is designed to improve the recommendation of tail items, the paper lacks a detailed analysis of how the optimization specifically impacts the proportion of long-tail items in the final recommendation results. Including this analysis would provide a clearer picture of the model's effectiveness in addressing the long-tail problem.
3. The use of Transformer-based models raises concerns about the computational complexity of the approach, particularly in large-scale recommendation systems. However, the paper does not address how the model’s complexity might affect scalability or the trade-offs between model performance and computational cost.
4.  There is a lack of sensitivity analysis regarding the hyperparameters in Equation (9), aside from the temperature factor.  A comprehensive evaluation of how these hyperparameters impact model performance would provide valuable insights into their roles and help in understanding their influence on the overall performance.

**Reviewer Confidence:**

3: The reviewer is confident but not certain that the evaluation is correct

**Scope:**

4: The work is relevant to the Web and to the track, and is of broad interest to the community

---

### Official Review · Reviewer_4Dcs · 2024-12-02

**Novelty:** 2
**Technical Quality:** 2

**Review:**

The paper addresses two critical challenges that lead to performance degradation in tail-item recommendations for sequential recommendation (SR) models utilizing large language models (LLMs) and large vision models (LVMs). The challenges are:
1. The "all-in ground truth" problem in cross-entropy loss.
2. The knowledge transfer tax.

To tackle these issues, the paper proposes R² Rec (i.e., _Reweighting_ and _Reembedding_), which initializes tail-item embeddings using a Gaussian distribution and incorporates a reweighting function to emphasize tail items. The authors validate their method through diverse experiments.

**Strengths**
- Addressing tail-item recommendations is an important contribution as it facilitates diversity while maintaining personalized recommendations.
- The attempt to integrate LLMs and LVMs into tail-item recommendations aligns well with recent trends.

**Weaknesses**:
The weaknesses are presented in the form of questions for further discussion.

&nbsp;

**Questions:**

**Q1. Validity and Novelty of Tail-Item Performance Degradation of LLMs/LVMs**

The paper asserts that LLM-/LVM-based models perform poorly on tail items. However, recent studies demonstrate that LLMs excel in zero-/few-shot scenarios [1], effectively addressing cold-start issues and even expanding into warm-start contexts [2].
Considering this trend, I question both the validity and novelty of the issue raised in the paper.

&nbsp;

**Q2. Lack of Specificity in Cross-Entropy Loss and Knowledge Transfer Tax**

**1. Cross-Entropy Loss:**
The "all-in ground truth" problem highlighted in the paper is not specific to LLMs/LVMs but is instead a general limitation of most recommendation models trained with cross-entropy loss. The paper should clarify how the structural characteristics of LLMs/LVMs exacerbate this issue.

**2. Knowledge Transfer Tax:**
The described phenomenon—dominant knowledge weakening the learning of tail items—appears to be an inherent issue of imbalanced item interactions rather than a unique problem of LLMs/LVMs.

**3. Terminology Suggestion:**
The term "Knowledge Transfer Tax" seems abstract. A more precise term, such as "Dominant Knowledge Overfitting," might better capture the described phenomenon.

&nbsp;

**Q3. Incomplete Organization of the Paper**

**1. Logical Flow:**
The logical flow of the paper feels somewhat disjointed due to three key issues:
(1) The significance of LLMs/LVMs in sequential recommendation is not clearly established.
(2) The importance of addressing long-tail recommendations is not well-motivated.
(3) The connection between the limitations of LLMs/LVMs and their impact on long-tail recommendation is unclear.


**2. Research Focus:**
(1) Is the paper primarily focused on long-tail recommendations? If so, is sequential recommendation necessary in this context?
(2) Does the paper aim to address large-scale knowledge from LLMs/LVMs, or is the focus on image-/text-based recommendation?

**3. Figure 1 (Right Part):**
The right part of Figure 1 does not convincingly show the limitations of LVMs. The observed degradations might simply result from embedding initialization rather than representing a fundamental limitation of LVMs. This claim feels overstated. Additionally, the paper should provide an explanation of why ID-based recommendation models perform relatively better in this context.

&nbsp;

**Q4. Lack of Technical Novelty in Proposed Methods**

The technical novelty of the proposed Reweighting and Reembedding methods is unclear.

**1. Reembedding**: Simply reinitializing tail-item embeddings using a Gaussian distribution.

**2. Reweighting**: Adjusting logit values using arbitrary hyperparameters.

Specific questions about the method:

1. Are the CLIP layers frozen or trainable?
2. In Equation (9), the signs of $a_p$ and $a_r$ seem incorrect. The paper states $+a_p, -a_r$, but shouldn’t it be $-a_p, +a_r$ to push tail items into the top-k? This is confusing; clarification would be helpful.

&nbsp;

**Minor Comments**
1. The paper’s title on OpenReview and the submitted manuscript are inconsistent. I recommend aligning them for consistency.
2. The long-tail item graph in Figure 1 (left part) looks different from the conventional long-tail graphs. Is there a specific reason for this?
3. Line numbers are only present on the first page. Adding line numbers throughout the paper would improve readability.
4. Ensure consistent capitalization for terms like "Top-K" and $\hat{i}^{j-1}_k$.
5. In Algorithm 1, line 15, consider using "(9)" instead of "9" to reference Equation (9).

&nbsp;
&nbsp;

[1] Hou et al., _Large Language Models are Zero-Shot Rankers for Recommender Systems_, ECIR'24.

[2] Kim et al., _Large Language Models meet Collaborative Filtering: An Efficient All-round LLM-based Recommender System_, KDD'24.

**Reviewer Confidence:**

4: The reviewer is certain that the evaluation is correct and very familiar with the relevant literature

**Scope:**

3: The work is somewhat relevant to the Web and to the track, and is of narrow interest to a sub-community

---

### Official Review · Reviewer_zmyV · 2024-12-02

**Novelty:** 5
**Technical Quality:** 4

**Review:**

Summary：
This paper proposes a reweighting and reembedding method for tail-item sequential recommendation. To address performance degradation issue for tail-item recommendation due to the assumption of standard cross-entropy loss, the authors propose to reinitialize tail item embedding and incorporate reweighting function to adjust the item weights during model training, enabling the model pays more attention to the tail items. Extensive experiments are conducted to validate the performance improvement on tail item recommendation.
Pros:
1 this is an interesting problem to explore the challenge of all-in ground-truth and knowledge transfer tax.
2 this paper analyzes the performance degradation of LLM/LVM-based recommendation models on tail items, and explores such issue from different aspects.
3 this paper proposes a reweighting function in the cross-entropy loss and adjusts the importance of the tail items during model training.
Cons:
1 In Sec3.2, the authors claim the deficiency of cross-entropy loss for the performance degradation from all-in ground-truth and knowledge transfer tax, respectively. 1）all-in ground-truth is an inherent flaw of cross-entropy loss; does this flaw only affect the recommendation performance of tail items? 2) What does Equation (3) aim to express or prove?
2 In Sec3.3, what does the notation f in eq (7) mean? How does the reweighting function work? What is the basis for these parameter settings in eq(9), e.g., \alph_{p}. additionally, Figure 2 provides a schematic of the method, but the main text does not explain the working process of each module using specific examples from the figure.
3 the CE loss treats all the items as equal, the authors argue that it is inappropriate due to the limited tail items, resulting insufficient training for such items, How can reinitialization training be performed to improve the tail item representations? This is not clearly explained.

**Questions:**

See reviews.

**Reviewer Confidence:**

3: The reviewer is confident but not certain that the evaluation is correct

**Scope:**

4: The work is relevant to the Web and to the track, and is of broad interest to the community

---

### Official Review · Reviewer_fJub · 2024-12-03

**Novelty:** 4
**Technical Quality:** 3

**Review:**

This paper addresses the challenge of poor performance of large vision models (LVMs) and large language models (LLMs) on tail items in sequential recommendation systems. The authors identify two main issues contributing to this performance degradation: (1) the "all-in groundtruth" problem, where standard cross-entropy loss focuses only on the target item and treats all non-target items equally, leading to insufficient optimization for tail items, and (2) the "knowledge transfer tax," where the external knowledge in LLMs and LVMs dominates the optimization process due to insufficient training on tail items.

Strengths

1. Novelty. The paper introduces a novel approach to address the long-standing issue of tail-item recommendation, which is a critical problem in the field of recommender systems.

2. Empirical Evidence. The authors provide extensive experimental results on three public datasets, demonstrating the effectiveness of R2Rec over multiple baselines, which strengthens the credibility of their approach.

3. Motivation. The paper specifically addresses the "all-in ground truth" and "knowledge transfer tax" issues, providing a targeted solution to improve the performance of LVMs and LLMs on tail items.

Weakness

1. Alternative Approaches. Simply discarding the VLM embeddings for long-tail items is a simple yet questionable approach. This weakened the motivation of the whole paper. It would be great to design a smoother approach to still use the VLM embedding for long-tail items, if possible. See the questions below for details.


2. Insufficient Analysis. There is limited analysis of the quality of the VLM embeddings for recommendation. Besides, the metrics used to measure the "insufficient learning" of embeddings are problematic. See the questions below for details.

**Questions:**

1. The effectiveness of the reweighting function on long-tail items makes me wonder to what extent the VLM embeddings are useful for recommendation. If the VLM embeddings for long-tail items are meaningless and can be discarded, then how about those items with middle frequency? To answer this question, it would be great if the authors could study the discriminativeness (such as the mutual information between these embeddings and the label. Quantization on embeddings may be needed to facilitate the calculation of MI) of items with different frequencies (high-middle-low).

2. Could the authors try other smoother approaches, such as a mixed representation (such as a weighted sum) of the VLM embedding and a re-initialized embedding? Simply discarding the VLM embeddings weakens the motivation to utilize VLM to enhance recommendation.

3. What's the purpose of the embedding visualization in Fig. 1 and Fig. 4? Are these blocks the raw embedding values or the norm of embeddings? If YES, then this visualization is questionable. For example, we can simply initialize the embeddings by N(1000, 1), and the values are large. There are several off-the-shelf methods to visualize the embeddings, such as the Quantized Mutual Information as mentioned before, to measure discriminativeness, the singular value spectrum [1,2] to measure the dimensional robustness, or the alignment and uniformity metrics [3, 4]. It would be great if the authors chose a more appropriate metric.

[1] Understanding Dimensional Collapse in Contrastive Self-Supervised Learning.

[2] On the Embedding Collapse When Scaling Up Recommendation Models.

[3] Understanding Contrastive Representation Learning through Alignment and Uniformity on the Hypersphere.

[4] Towards Representation Alignment and Uniformity in Collaborative Filtering.

**Reviewer Confidence:**

4: The reviewer is certain that the evaluation is correct and very familiar with the relevant literature

**Scope:**

4: The work is relevant to the Web and to the track, and is of broad interest to the community